**PLOS** NEGLECTED TROPICAL DISEASES

# Epidemiological and clinical features of hump-nosed pit viper (*Hypnale hypnale* and *Hypnale zara*) envenoming in children

**R. M. M. K. Namal Rathnayaka**[1,2,3]*, **P. E. Anusha Nishanthi Ranathunga**[4], **S. A. M. Kularatne**[5]

**1** Department of Pharmacology, Faculty of Medicine, Sabaragamuwa University of Sri Lanka, Hidellana, Ratnapura, Sri Lanka, **2** Department of Veterinary Pathobiology, Faculty of Veterinary Medicine and Animal Science, University of Peradeniya, Sri Lanka, **3** Intensive care unit, Teaching Hospital, Ratnapura, Sri Lanka, **4** Medical Unit, Teaching Hospital, Ratnapura, Sri Lanka, **5** Department of Medicine, Faculty of Medicine, University of Peradeniya, Sri Lanka

* namal@med.sab.ac.lk

## Abstract

**Data Availability Statement:** All relevant data are within the manuscript and its Supporting Information files.

### Background

Bites by the hump-nosed pit vipers (HNPV) of the genus *Hypnale* are the commonest type of venomous snakebites in Sri Lanka. Their bites frequently cause local effects while rarely causing systemic envenoming, that may include acute kidney injury and coagulopathy. There are 3 species of genus *Hypnale* including *H. hypnale*, *H. zara* and *H. nepa* from which latter two are endemic to Sri Lanka. Virtually all studies on HNPV bites in Sri Lanka are focused on adults except two studies in paediatric group. The aims of this study were to describe the epidemiology and clinical manifestations of HNPV bites in a group of children admitted to a tertiary care hospital in Sri Lanka.

### Methodology/Principal findings

This was a prospective observational study carried out in Teaching Hospital Ratnapura, Sri Lanka over 27 months commencing from May 2020 including all children aged up to 14 years with the history of HNPV bites.

There were 40 (56%) HNPV bites, of them 28 (70%) were males. The age was 84 months (50.2–120 months). Majority (n = 21;52.5%) were bitten during day-time (06:00–17:59) in home gardens (n = 20; 50%) on lower limbs (n = 24;60%). Most children (n = 30;75%) were admitted to the medical facility < 4 hours after the snakebite [90 min (40–210 min)] and the hospital stay was 4 days (3–5 days). Local envenoming was observed in 38 patients (95%) and systemic effects developed in 4 patients (10%) as mild coagulopathy. Local effects include local pain (n = 30; 94%), swelling (n = 38;95%), blistering (n = 11;27.5%), necrosis at the site of bite (n = 11; 27.5%), regional lymph node enlargement (n = 8;20%) and local bleeding (n = 4;10%). For the local effects, surgical interventions were needed in 10 children (25%) and 3 (7.5%) of them developed acute compartment syndrome leading to fasciotomy. Leucocytosis (n = 28;78%) and eosinophilia (n = 9;27%)

**Funding:** The authors received no specific funding for this work.

**Competing interests:** The authors declare no competing interests.

were the prominent laboratory findings. All got recovered except in patients with fasciotomy who got permanent scar.

## Conclusions/Significance

Hump-nosed pit viper bites mostly cause local effects and rarely systemic envenoming in children. Compartment syndrome is common in children following their bites.

### Author summary

Snakebite is a neglected tropical disease affecting mostly rural communities who are engaging in agricultural activities. Many studies focus on adult snakebites and very few are describing on snakebites in paediatric group. Therefore, the understanding of epidemiological and clinical profile of snakebites in children is lacking and poorly characterized. In Sri Lanka, hump-nosed pit viper (*Hypnale spp.*) causes the commonest venomous snakebites because it inhabits all over the country and is found frequently in human habitat. We undertook a clinical study in Teaching Hospital Ratnapura in order to describe clinical and epidemiological characteristics of HNPV bites in children.

There were 40 (56%) children with HNPV bites from which 95% (n = 38) developed local envenoming such as local pain, swelling, blistering and necrosis at the site of bite. Systemic envenoming was observed in 4 (10%) who had mild coagulopathy and 3 (7.5%) had non-specific envenoming features including abdominal pain and headache. The prominent finding was the occurrence of compartment syndrome in some children (n = 3;7.5%) who needed to open limb compartment (fasciotomy).

## Introduction

Bites by the hump-nosed pit vipers (HNPV) of the genus *Hypnale* are the commonest type of venomous snakebites in Sri Lanka causing 22–77% of all snakebites [1] in both adults [2–5] and children [6,7]. They inhabit all over Sri Lanka and small region (Western Ghats region) of India. There are 3 species of genus *Hypnale* including *H. hypnale*, *H. zara* and *H. nepa* from which latter two are endemic to Sri Lanka [8]. Out of 3 species, majority of bites are caused by *H. hypnale* (80–82%), then *H. zara* (14–22%) and *H. nepa* (4–5%) [3–5]. This different biting frequency is due to the geographical distribution of 3 species in the country. Despite large number of studies in adults on HNPV bites in Sri Lanka [2–5,9], only two paediatric studies are found in literature [6,7]. In children, like in adults, bites mostly cause local effects such as local pain, swelling, blistering, necrosis and lymphadenopathy [6,7]. The systemic envenoming effects are mainly acute kidney injury (AKI) and venom induced consumption coagulopathy (VICC) [2–4,9–11]. In addition, thrombotic microangiopathy (TMA), chronic kidney disease, chronic wounds [2,9,10] and cardiac complications [12] may rarely occur.

Snakebite is a preventable life-threatening medical accident and the envenoming effects are more in children because of higher ratio of injected venom to the body mass. Also, the risk of snakebite is more in children because of their innate curiosity to know about creatures, meddling in snake habitat and barefoot walking. The aims of this study were to describe epidemiology and clinical manifestations of HNPV bites in a group of children admitted over 27 months to a tertiary care hospital in Sri Lanka.

## Methods

### Ethics statement

Ethical approval for the study was obtained from the Faculty of Medicine, University of Peradeniya (2020/EC/58). Informed written consent was obtained from parents or guardian of each participant before collecting data. Informed consent was also obtained from participants where applicable. Further, informed consent was obtained from the mother of serial No. 21 patient in order to publish the images. Authors confirm that all methods in this study were carried out in accordance with relevant hospital guidelines and regulations.

### Study design and setting

This was a prospective observational clinical study, carried out in paediatric wards, Teaching Hospital Ratnapura, Sri Lanka over 27 months commencing from May 2020 that included all children with HNPV bites. Up to 14 year of age (including 14 year) was considered as the paediatric age group. On admission, patients were assessed by the principal investigator and reassessed daily until hospital discharge. Epidemiological features, clinical manifestations, laboratory findings, treatments and short-term outcomes were recorded in a formatted data sheet. Data were collected using an interviewer-administered questionnaire. The species of live or dead specimen of offending snakes were identified using a standard key [8] by the principal investigator.

Local envenoming such as pain, swelling, bleeding, necrosis, and lymph node enlargement were evaluated. The severity of local pain was categorized as mild (presence of pain recalled by direct questioning), moderate (pain is severe enough to produce discomfort and crying in palpation or not allow to palpate the bitten limb) and severe (crying with sleep disturbance). Similarly, observations of local swelling were graded as mild, moderate or severe depending on the extent of the swelling and the severity of the involvement. Swelling confined only to the site of bite was graded mild; extension to more than half of the limb was graded moderate; and extension to whole limb was graded severe. Coagulopathy was assessed using bleeding manifestations, 20 min whole blood clotting test (WBCT20) and clotting profile (PT/INR, aPTT). All patients were followed for the detection of consumption coagulopathy at 6-hour intervals of WBCT20 for 1–2 days and monitoring of any bleeding manifestation. On admission, clotting profile and WBCT20 were performed. When positive findings were detected clotting profile was done daily for some patients. Kidney injury was assessed using urine output, blood urea and serum creatinine. Laboratory assessment included PT/INR, aPTT, complete blood count (white blood cells, neutrophils, lymphocytes, eosinophils, platelets, haemoglobin), red cell indices (mean corpuscular volume-MCV, mean corpuscular haemoglobin-MCH, mean corpuscular heamoglobin concentration-MCHC), blood urea, serum creatinine, and serum electrolytes. The morphological characteristics of snakes (gender, head length, tail length, snout to vent length, total length and scale counts) were recorded and dead specimens were preserved in 10% formalin and labeled with patient's serial number and the date of admission. They were deposited at Teaching Hospital Ratnapura for proof and live snakes were released to their natural habit. Data analysis was done using SPSS version 21 and they were presented as median and interquartile range (IQR).

## Results

### Identification of snakes

There were 14 (35%) killed specimens (*H. hypnale*-13 and *H. zara*-1) and 12 (30%) live snakes (*H. hypnale*-8 and *H. zara*-4, Fig 1) in current study.

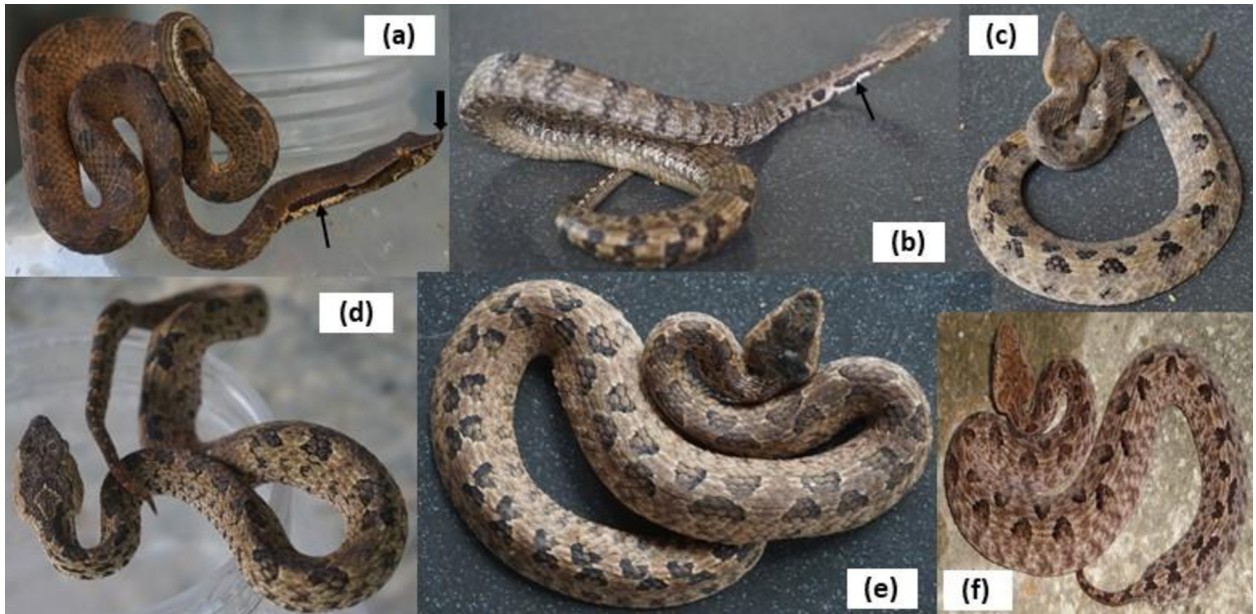

**Fig 1. Live hump-nosed pit vipers brought by the patients.** (a), (b) *Hypnale zara* (thick black arrow indicates the prominent hump and thin black arrow indicates the characteristic nuchal colour band (c), (d), (e), (f) *Hypnale hypnale* (hump is not prominent).

## Epidemiological features

During the study period, there were 72 snakebite admissions to the paediatric wards in Teaching Hospital Ratnapura from which, HNPV bites were 40 (56%). Out of these, 21 (52.5%) were bitten by *H. hypnale* and 5 (12.5%) were bitten by *H. zara*. Other 14 (35%) were bitten by *Hypnale spp.* and their exact species have not been identified because these patients were transferred from local hospitals in which offending snakes were identified only as hump-nosed pit vipers by the admitting medical officers. The demographic and epidemiological features are shown in Table 1.

Males (n = 28;70%) outnumbered females. The median age was 84 months [50.2–120 months] and age ranged between 5–167 months. Majority (n = 19; 47.5%) were in 5–10 years of age category. Most children (n = 21;52.5%) were bitten at day-time (06:00–17:59) in home gardens (n = 20; 50%). Lower limbs (24;60%) were affected more than upper limbs and majority (n = 13; 32.5%) were bitten on feet. There were 3 (7.5%) patients with multiple sites of bite. A 7-year-old girl (Serial No.21) was bitten by *H. zara* on face, dorsum of right hand, 3rd finger and right knee joint while she was sleeping on a bed. She had 7 fang punctures (Fig 2).

*Hypnale* bites occurred throughout the year. But majority of them occurred on October (n = 9; 22.5%), then on July and December (each n = 6;15%) (Fig 3).

Most of children (n = 19;47.5%) were in grade 1–5. Most bites (n = 26; 65%) occurred without provocation. Majority of patients were from Ratnapura administrative division (n = 15;37.5%) and then in Kuruvita (9;22.5%). Most children (n = 30;75%) were admitted to the medical facility < 4 hours after the snakebite [90 min (40–210 min)] and 12 (30%) were admitted < 1 hour. Hospital stay was 4 days (3–5 days) and the range of hospital stay was 2–18 days. The commonest first aid method was washing the bite site (n = 26;65%). Thirteen patients (32.5%) were transferred from local hospitals.

**Table 1. Demographic and epidemiological features of hump-nosed pit viper (*H. Hypnale* and *H. zara*) bites in children.**

| Demographic and epidemiological features | Number (%) | Demographic and epidemiological features | Number (%) |
|---|---|---|---|
| **Gender** | | **Provocation** | |
| Male | 28 (70) | Yes | 3 (7.5) |
| Female | 12 (30) | No | 26 (65) |
| | | Not decided | 11 (27.5) |
| **Age range** | 5 months-13 years & 11 months | **First aid measures** | |
| < 1 year | 1 (2.5) | Not applied | 13 (32.5) |
| 1–5 year | 13 (32.5) | Applied | 27 (67.5) |
| 5 years & 1 month-10 year | 19 (47.5) | Washing | 26 (65) |
| > 10 years | 7 (17.5) | Ligation | 12 (30) |
| | | Both washing & ligation | 11 (27.5) |
| Time of bite | | **Native treatment** | |
| Daytime bites (00:06–17:59) | 21 (52.5) | No | 34 (85) |
| 00:06–11:59 | 6 (15) | Yes | 6 (15) |
| 12:00–17:59 | 15 (37.5) | Ingestion of curry leaf (*Murraya koenigii*) broth | 6 (15) |
| Night-time bites (18:00–05:59) | 19 (47.5) | | |
| 18:00–11:59 | 16 (40) | | |
| 00:00–05:59 | 3 (7.5) | | |
| **Site of bites** | | **Duration of hospital stay** (days) | |
| Lower limbs | 24 (60) | 2 | 5 (12.5) |
| Feet | 13 (32.5) | 3 | 14 (35) |
| Toes | 4 (10) | 4 | 7 (17.5) |
| Great toes | 4 (10) | 5 | 6 (15) |
| Legs | 2 (5) | 6 | 1 (2.5) |
| Ankles | 1 (2.5) | 7 | 0 |
| Upper limbs | 13 (32.5) | 8 | 2 (5) |
| Fingers | 11 (27.5) | > 8 | 3 (7.5) |
| Thumbs | 1 (2.5) | Left against medical advice | 2 (5) |
| Hands | 1 (2.5) | | |
| Multiple sites | 3 (7.5) | | |
| **Place of bites** | | **Previous snakebite** | |
| Home garden | 20 (50) | No | 39 (97.5) |
| Inside home | 9 (22.5) | Yes | 1 (2.5) |
| Foot path | 3 (7.5) | | |
| Concrete road | 3 (7.5) | | |
| Estate (tea and rubber) | 2 (5) | | |
| On bicycle | 2 (5) | | |
| Tarry road | 1 (2.5) | | |
| **Time to hospital admittance** (h) | | **Previous/current medical conditions** | |
| < 1 | 12 (30) | No | 35 (87.5) |
| 1–1.59 | 9 (22.5) | Yes | 5 (12.5) |
| 2–3.59 | 9 (22.5) | Febrile convulsions | 2 (5) |
| 4–5.59 | 7 (17.5) | Asthma | 2 (5) |
| 6–7.59 | 2 (5) | Acute lymphoblastic leukemia | 1 (2.5) |
| > 8 | 1 (2.5) | | |
| **Education** | | **Hospital admissions** | |
| No (not applicable) | 11 (27.5) | Direct admissions | 27 (67.5) |
| Nursery | 2 (5) | Transfers | 13 (32.5) |
| Grade 1–5 | 19 (47.5) | | |
| Grade 6–9 | 8 (20) | | |

## Clinical features

Clinical manifestations of HNPV bites in children are shown in Table 2 and Fig 4. There were 2 (5%) patients with dry bites who did not show any envenoming features in spite of having fang punctures at the site of bite with the availability of the specimen of offending snake. Local envenoming was observed in 38 (95%) children. Systemic manifestations were observed in 4 (10%) who had VICC. Three (7.5%) were found to have non-specific envenoming features

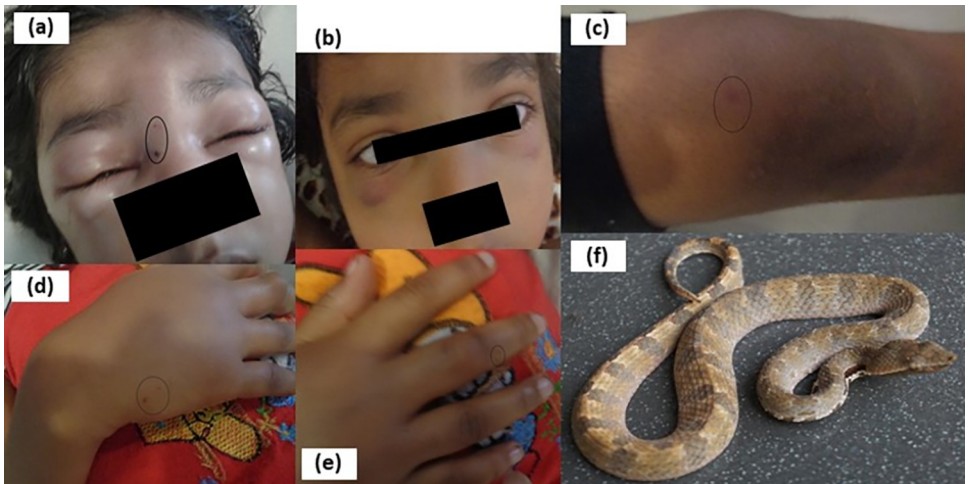

**Fig 2. Multiple sites of bite in S.No.21 patient.** (a) 2 fang punctures on face (circled) on day 1 (b) full recovery on day 4 (c) fang puncture (circled) just above right knee joint (d) 2 punctures (circled) on right hand (e) 2 fang punctures (circled) over right 3ʳᵈ finger (f) offending live hump-nosed pit viper: *Hypnale zara* from Kuruvita, Sri Lanka (06˚77'N, 80˚37'elevation 37m/ 121ft).

including abdominal pain in 2 (5%) and headache in 1 (2.5%). Fang punctures were observed in all patients from which, majority had 1 puncture (n = 18;45%).

Pain assessment was difficult in children below the age of 3 years which included 8 (20%). Therefore, pain assessment was done in 32 whose age was more than 3 years. Out of 32, mild

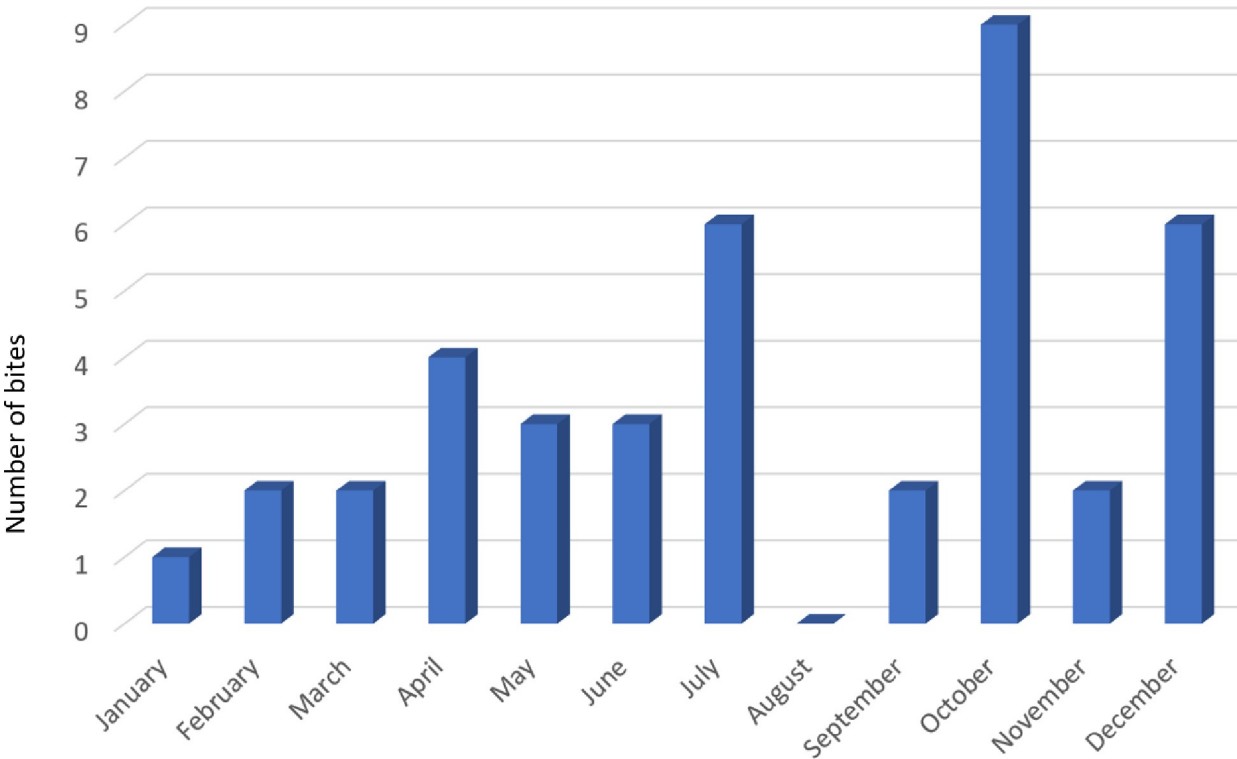

**Fig 3. Monthly distribution of hump-nosed pit viper bites in children.**

**Table 2. Clinical features of hump-nosed pit viper (Genus: *Hypnale*) bites in children.**

| Clinical feature | Number (%) |
|---|---|
| Dry bites | 2 (5) |
| **Local envenoming** | 38 (95) |
| Pain | 30 (94) |
| Mild | 5 (16) |
| Moderate | 10 (31) |
| Severe | 15 (47) |
| Swelling | 38 (95) |
| Mild | 3 (7.5) |
| Moderate | 18 (45) |
| Severe | 17 (42.5) |
| Blistering | 11 (27.5) |
| Necrosis at the site of bite | 11 (27.5) |
| Lymph node enlargement | 8 (20) |
| Local bleeding | 4 (10) |
| Bruising | 3 (7.5) |
| Fang punctures | 40 (100) |
| 1 puncture | 18 (45) |
| 2 punctures | 17 (42.5) |
| 3 punctures | 4 (10) |
| 7 punctures in multiple sites | 1 (2.5) |
| **Nonspecific features** | 3 (8) |
| Abdominal pain | 2 (5) |
| Headache | 1 (3) |
| **Systemic manifestations** | |
| Venom induced consumption coagulopathy | 4 (10) |
| **Complications** | |
| Myalgia | 8 (20) |
| Cellulitis | 5 (12.5) |
| Compartment syndrome leading to fasciotomy | 3 (7.5) |
| **Treatment** | |
| Surgical intervention | 10 (25) |
| Wound cleansing | 6 (15) |
| Incision and drainage | 2 (5) |
| Skin grafting | 2 (5) |
| Secondary suturing | 2 (5) |
| **Outcomes** | |
| Fully recovered | 38 (95) |
| Left against medical advice | 2 (5) |

pain was observed in 5 (16%), moderate pain in 10 (31%) and severe pain in 15 (47%). No pain was observed (dry bites) in 2 children (6%). Compartment syndrome was detected in 3 (7.5%) children who required fasciotomy.

 Laboratory findings

 Laboratory findings are shown in Tables 3 and 4. Out of haematological parameters, majority (n = 28;78%) had leukocytosis (elevated white blood cell counts, normal 4-11x $10^3$/μL) and eosinophilia (elevated eosinophils > 500 /μL) was observed in 9 (27%) children. Mild elevation of liver enzymes (3 times the normal) were detected in 7 (29%) patients.

## Treatment

No antivenom is currently available in Sri Lanka or India for HNPV envenoming. Therefore, supportive treatment was carried out for all 40 children with HNPV bites. Paracetamol was given 6 hourly or thrice a day for pain management. Local swelling was managed with elevation of the affected limb and close monitoring of peripheral pulse, $SpO_2$ and capillary refilling time (CRFT) for the detection of acute compartment syndrome. If cellulitis was suspected, oral or intravenous cloxacillin 6 hourly and metronidazole 8 hourly were administered. Fluid

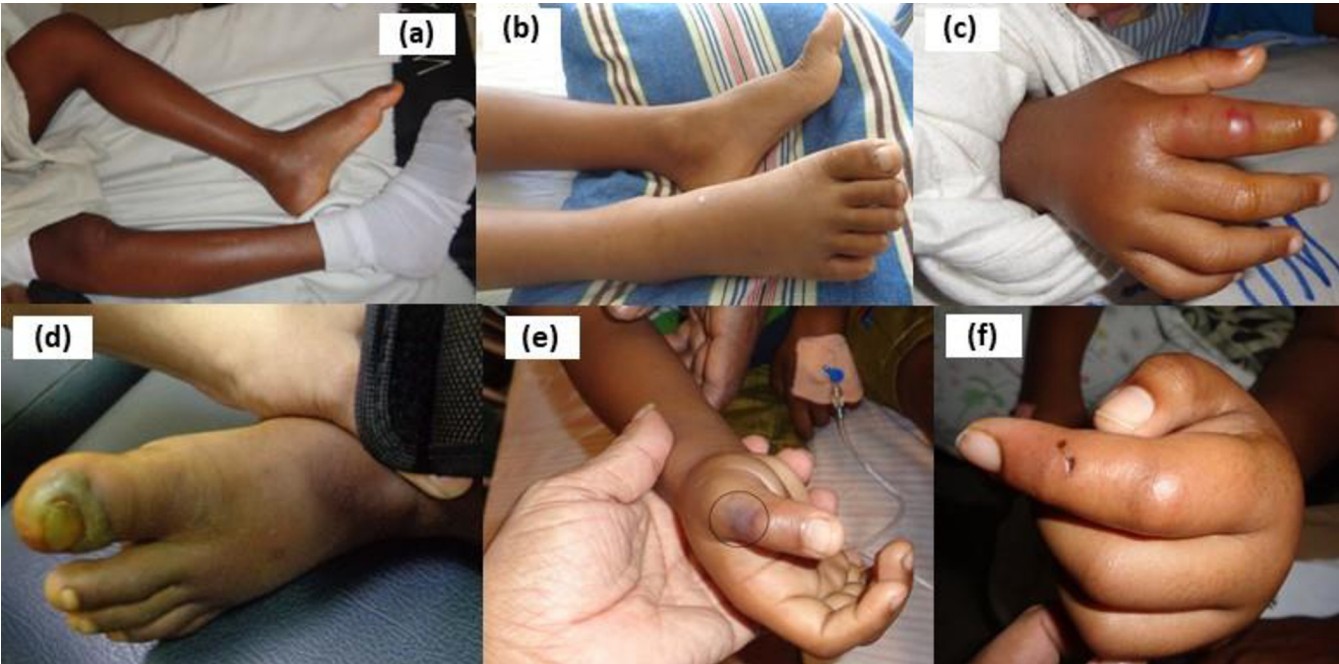

**Fig 4. Local envenoming manifestations of hump-nosed pit viper bites in children.** (a) severe local swelling of left lower limb on day 2 (b) moderate local swelling of right lower limb on day 2 (c) haemorrhagic blister on right 2nd finger on day 3 (d) heamorrhagic blister on left big toe on day 2 (e) necrosis of right thumb (circled) on day 2 (f) bite marks on left 2nd finger on day 2.

input-output was monitored with blood urea and serum creatinine levels for assessment of renal function.

Surgical interventions were done for 10 (25%) children including wound cleansing, incision and drainage, skin grafting, fasciotomy and secondary suturing. Fasciotomy was done in 3 (7.5%) children as described follows.

**Table 3. Laboratory findings of hump-nosed pit viper envenoming in children.**

| Laboratory finding | Number (%) | Range of abnormal lab findings | Normal range |
|---|---|---|---|
| **Haematological parameters** | | | |
| Leukocytosis | 28 (78) | $11–23 \times 10^3/\mu L$ | $4–10 \times 10^3/\mu L$ |
| Neutrophil leukocytosis | 19 (58) | $7.52–17.8 \times 10^3/\mu L$ | $2–7 \times 103/\mu L$ |
| Lymphocytosis | 13 (39) | $4.03–9.68 \times 10^3/\mu L$ | $0.8–4 \times 10^3/\mu L$ |
| Eosinophilia | 9 (27) | $0.51–1.36 \times 10^3/\mu L$ | $0.02–0.5 \times 10^3/\mu L$ |
| Thrombocytopenia | 1 (3) | $131 \times 10^3/\mu L$ | $150–450 \times 10^3/\mu L$ |
| Decreased haemoglobin | 6 (17) | 10.3–10.8 g/dL | 11–16 g/dL |
| **Red cell indices** | | | |
| Decreased MCV | 20 (59) | 61–79.1 fL | 80–100 fL |
| Decreased MCH | 20 (59) | 18.4–26.6 pg | 27–34 pg |
| Decreased MCHC | 2 (6) | 29.5–30.1 g/dL | 32–36 g/dL |
| Decreased MPV | 1 (3) | 5.8 fL | 6.5–12 Fl |
| **Clotting profile** | | | |
| Positive 20 min WBCT | 2 (5) | > 20 min | < 20 min |
| Elevated PT | 4 (14) | 18–19.2 s | 10–15 s |
| Elevated INR | 4 (14) | 1.48–1.62 | 1–1.4 |
| Elevated aPTT | 12 (52) | 31–43 s | 25–30 s |
| **Biochemical parameters** | | | |
| Elevated SGOT/AST (*mild) | 7 (29) | 36–107 U/I | 10–35 U/I |
| Elevated SGPT/ALT (*mild) | 2 (9) | 43–88 U/I | 10–40 U/I |

*mild-3 times normal

**Table 4. Descriptive statistics of laboratory findings in hump-nosed pit viper envenoming in children.**

| Laboratory finding | Number | Range | Minimum | maximum | Mean | Std. Deviation |
|---|---|---|---|---|---|---|
| WBC count (x $10^3$/μL) | 36 | 15 | 8 | 23 | 13.9 | 4.2 |
| Neutrophil % | 36 | 69 | 22 | 91 | 62.7 | 18.3 |
| Neutrophil count (x $10^3$/μL) | 33 | 16678 | 1122 | 17800 | 8042.2 | 4004.5 |
| Lymphocyte % | 36 | 62 | 5 | 67 | 29.8 | 16.4 |
| Lymphocyte count (x $10^3$/μL) | 33 | 8630 | 1050 | 9680 | 3790.6 | 2321.3 |
| Eosinophil % | 32 | 10.3 | .1 | 10.4 | 2.4 | 2.6 |
| Eosinophil count (x $10^3$/μL) | 33 | 1356 | 5 | 1360 | 302.9 | 334.6 |
| Hb (g/dL) | 36 | 3.5 | 10.2 | 13.7 | 12.3 | 1.1 |
| Platelet count (x $10^3$/μL) | 36 | 598 | 131 | 729 | 361.8 | 100.4 |
| PT (s) | 27 | 7.9 | 11.3 | 19.2 | 14.4 | 2.1 |
| INR | 28 | .68 | .94 | 1.62 | 1.2 | 0.2 |
| aPTT (s) | 23 | 20 | 23 | 43 | 33.2 | 5.7 |
| creatinine (μmol/L) | 35 | 62 | 24 | 86 | 45.8 | 13.2 |
| Blood urea (mmol/L) | 35 | 4.1 | 1.2 | 5.3 | 3.1 | 0.96 |
| Na (mmol/L) | 34 | 11 | 132 | 143 | 139.1 | 2.3 |
| K (mmol/L) | 34 | 1.8 | 3.6 | 5.4 | 4.3 | 0.4 |
| SGOT/AST (U/I) | 24 | 92 | 15 | 107 | 35.0 | 16.8 |
| SGPT/ALT (U/I) | 23 | 75 | 13 | 88 | 22.2 | 15.7 |
| CRP (mg/L) | 24 | 13 | 5 | 18 | 6.1 | 3.1 |
| MCV (fL) | 34 | 27.0 | 61.0 | 88.0 | 78.0 | 5.3 |
| MCH (pg) | 34 | 11.8 | 18.4 | 30.2 | 26.2 | 2.2 |
| MCHC (g/dL) | 34 | 9.0 | 29.5 | 38.5 | 33.8 | 1.9 |
| MPV (fL) | 34 | 4.4 | 5.8 | 10.2 | 8.5 | 0.8 |

## Case 1 (Serial No.33 patient)

A 4-year-old girl was admitted to Emergency Treatment Unit following HNPV bite to her medial aspect of right foot and the snake was identified as *H. hypnale* (Fig 5a and 5b) by the principal investigator. She was bitten while she was walking to home on a foot pathway at about 1020 h. Initially, site of bite was washed with soap and then a tourniquet was applied above the fang punctures for 30 minutes. On admission, the child had severe local pain over right lower limb with moderate swelling. On examination, she had 3 bite marks and there was no blistering or necrosis. Her WBCT20 on admission was normal and the other laboratory findings are shown in Table 5. She was kept on close observation by monitoring CRFT, dorsalis pedis pulse and $O_2$ saturation ($SpO_2$) of the affected limb. Gradually, her local swelling increased with prolonging CRFT (Fig 5c and 5d) associated with low $SpO_2$. She was then undergone right lower limb fasciotomy on day 2 of snakebite. Skin grafting was done on day 15 of snakebite and was discharged on day 18 with the arrangement of clinic visit in surgical unit.

## Case 2 (Serial No.39 patient)

A 3-year-old previously healthy boy was transferred from a local hospital at 1205 h following HNPV bite to dorsum of his right hand at 1630 h previous day while he was playing inside home. On admission to the tertiary care centre, he had severe pain over the bitten arm and his urine output was normal. On examination, there were 2 fang punctures on right hand and no local bleeding or necrosis was observed. Capillary refilling time of affected arm was < 2 sec and $O_2$ saturation was normal. He was kept on arm elevated and monitoring of CRFT with $SpO_2$. His laboratory findings are shown in Table 5. He developed lowering of $SpO_2$ with weak radial pulse suggestive of right upper limb acute compartment syndrome for which forearm fasciotomy was done on day 2 of snakebite. On and off, he was undergone wound debridement and skin grafting was done on day 14 of snakebite. He was discharged on day 16 with arranging clinic visits.

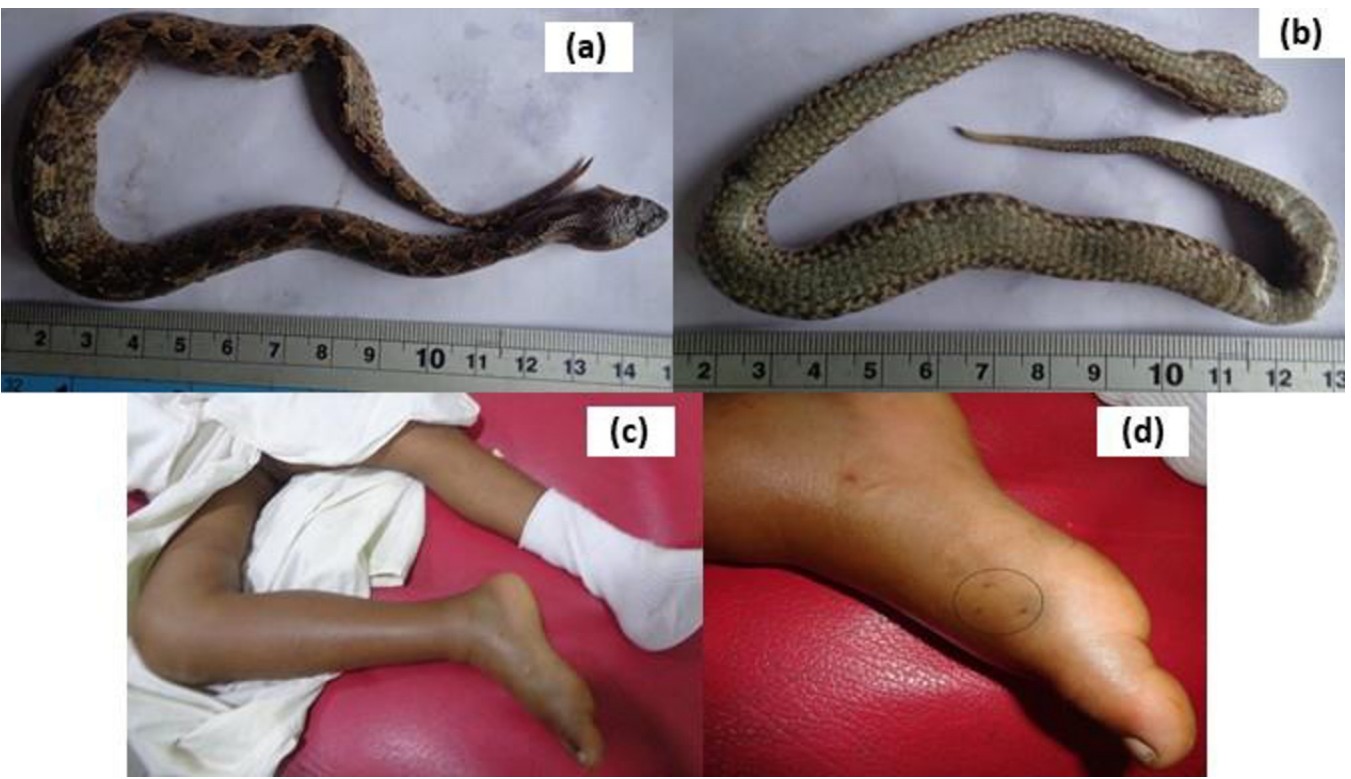

**Fig 5. Species of snake responsible for bite in case 1 patient-*H. hypnale*.** (a) dorsal aspect (b) ventral aspect (c) severe swelling of right lower limb (d) 3 fang punctures (circled) of the medial side of right foot.

**Table 5. Laboratory findings of patients who underwent fasciotomy.**

| Laboratory finding | Patient1 | | | Patient 2 | | | Patient 3 | | |
|---|---|---|---|---|---|---|---|---|---|
| | Day 1 | Day 2 | Day 3 | Day 1 | Day 2 | Day 3 | Day 1 | Day 2 | Day 3 |
| WBC (x10³/μL) | 9 | 7 | 7 | 16 | 18 | 13 | 11 | 9 | 9 |
| Neutrophils (%) | 81 | 52 | 38 | 47 | 42 | 37 | 82 | 59 | 37 |
| Neutrophil count (/μL) | 6870 | 3750 | 2600 | 6910 | 4110 | 4190 | 8810 | 5300 | 3270 |
| Platelets (x10³/μL) | 301 | 167 | 280 | 287 | 350 | 531 | 411 | 399 | 321 |
| Hb (g/dL) | 13.1 | 10.9 | 12.3 | 11.7 | 12.5 | 12.3 | 10.8 | 10.6 | 12.8 |
| PT (sec) | 13/12 | 14.8/12 | 15/12 | 11.8/12 | 13.2/12 | 12.4/12 | 14.5/12 | 13/12 | |
| INR | 1.09 | 1.28 | 1.29 | 0.98 | 1.1 | 1.03 | 1.21 | 1.08 | |
| aPTT (sec) | 31/25 | 37/25 | 34/25 | 42.6/32 | 37.4/32 | 35/32 | 28/30 | 21/30 | |
| Na⁺ (mmol/L) | 141 | 140 | | 138 | 142 | 141 | 142 | 145 | 136 |
| K⁺(mmol/L) | 4.3 | 4.1 | | 4.3 | 4.1 | 4.3 | 4.1 | 4.7 | 4.5 |
| Blood urea (mmol/L) | 2.3 | 2.6 | | 3.6 | 2.5 | 4 | 3 | 1.3 | |
| Creatinine (μmol/L) | 37 | 47 | | 47 | 34 | 47 | 49 | 38 | 45 |
| SGOT(AST) [U/I] | 34 | | | 49 | | | 41 | | |
| SGPT(ALT) [U/I] | 20 | | | 16 | | | 14 | | |
| CRP (mg/L) | < 5 | | | < 5 | | < 5 | < 5 | < 6 | < 6 |
| WBCT20 (min) | < 20 | < 20 | | < 20 | | | < 20 | | |

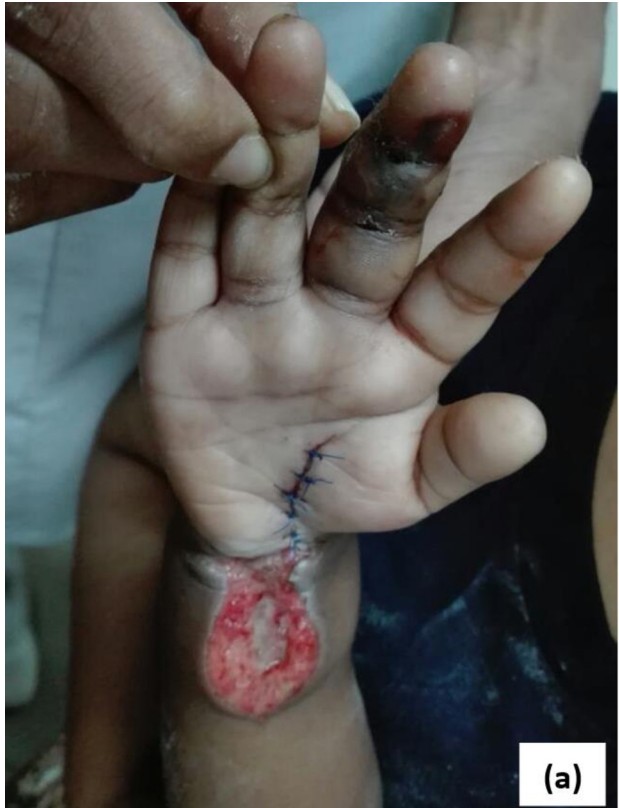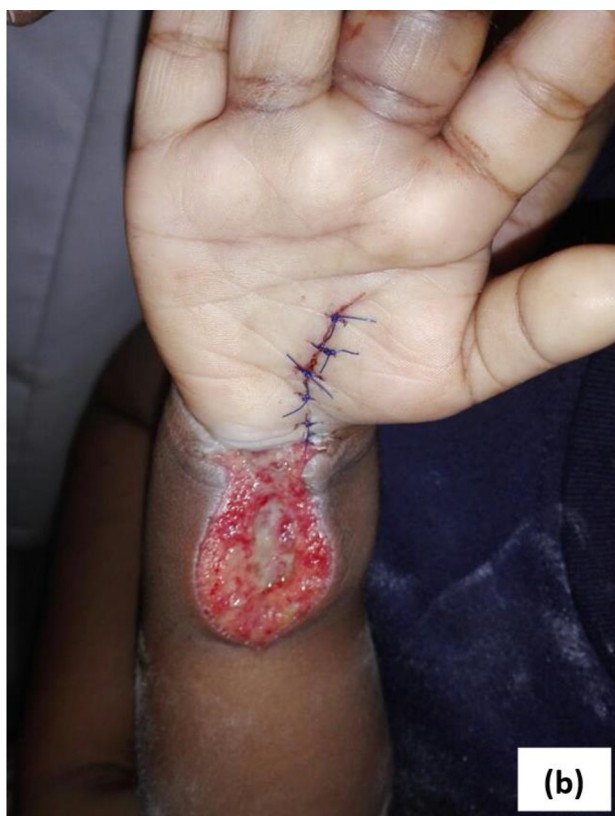

**Fig 6.** Carpel tunnel decompression and forearm fasciotomy of right hand in case 3 patient (a), (b) Note that the necrosis of the site of bite in middle finger and secondary suturing.

### Case 3 (Serial No.52 patient)

A 2 ½-year-old previously healthy boy was transferred from a local hospital following HNPV bite to his right middle finger at 2400 h while he was trying to get on to his bicycle. At that time, the snake was on the bicycle seat. Within 45 min, child was admitted to the local hospital. On admission to tertiary care centre, there was severe local pain and severe swelling over the bitten upper limb and 2 fang punctures were observed. There was a haemorrhagic blister, necrosis at the site of bite and local lymphadenopathy. Twenty minutes whole blood clotting test on admission was normal and the other laboratory findings are shown in Table 5. He was kept under close observation of CRFT, radial pulse and $SpO_2$ of the affected hand. As local swelling was gradually increased with lowering of $O_2$ saturation, right side carpel tunnel decompression and forearm fasciotomy was done under general anaesthesia on day 1 of snakebite (Fig 6). Then secondary suturing was done at day 3 and day 13. He was discharged on day 14 of snakebite with arranging clinic visits.

### Discussion

Current study is the first study done involving large number of children with HNPV bites (n = 40) compared to two previous studies in Sri Lanka [6,7]. This study points out that similar to adults [2–5,13], local effects are common and systemic effects are rare following HNPV bites in children. Like in adults, HNPV bites are the commonest cause of snakebites in children which is 44% [7]. In current study, it was 56% of all snakebites. A previous study describes that

38.5% of victims (n = 5) developed systemic envenoming including 5 children having VICC from which 2 had AKI [6]. Therefore, spectrum of systemic envenoming is similar in both adults and children in HNPV bites. Recently published case report describes an 8-year-old girl developed VICC, hemolytic uremic syndrome and acute hepatic injury following a HNPV bite and she got full recovery being treated 32 days in hospital [14]. We also found 4 children (10%) developing VICC as systemic effects with mild elevation of clotting profile and positive WBCT20 without having clinically detectable bleeding and even this abnormality prevailed only for one day. In children with HNPV bites, VICC occurs in 28.6% and AKI in 14.3% [6] whereas in adults, VICC occurs in 3.5–39% [2–5,11] and AKI in 6–10% [2,4,10]. However, AKI was not observed in present study. Thus, severe systemic effects were not observed in current study. This may be due to composition of venom variation according to the geographical region. However, renal failure complicating snakebite is more frequent and more severe in children than in adults [15].

Like in previous studies [16–18], current study also highlights the male predominance. Boys have higher rates of snakebites as they are more prone to catch, kill or interfere with snakes [19]. There is a difference in hospital stay between children and adults following HNPV bites. In current study, hospital stay was 4 days (3–5 days). But in adults, it was 2.5 days (2–3 days) [4]. This means children are in longer hospitalization than adults following HNPV bites. One reason for this could be due to the difficulty in clinical assessment of younger children below the age of 3 years.

We found acute compartment syndrome is higher in children, needing fasciotomy as the treatment option in 7.5% (n = 3). These children needed prolonged hospitalization (14–18 days) for the wound which needed secondary suturing and/or skin grafting later. This was actually a severe burden to the family because one of parents, usually mother had to stay with the child throughout the hospital stay. It was found that the fasciotomy is done 0.7% of paediatric snakebites [20]. Acute compartment syndrome is a medical emergency for which the treatment is surgery to open the compartment (fasciotomy). It is an elevation of intra-compartmental pressure to a level that impairs circulation. The sequelae of a delayed diagnosis of compartment syndrome is devastating. Symptoms of acute compartment syndrome include severe pain, elevation of CRFT, weak peripheral pulses, decreased ability to move or a pale color of the affected limb. Diagnosis is usually clinical or by measurement of intra-compartmental pressure. However, in snakebites, distal pulselessness and pallor may be explained by circulatory shock and local edema owing to toxic effects at the injection site and not necessarily by the development of compartment syndrome. Therefore, sometimes unnecessary fasciotomy may lead to severe disability [21]. Thus, it is suggested to get serial compartment pressure with the evidence of neurovascular compromise that is consistent with tissue damage [22]. In Sri Lankan hospital settings, compartment syndrome is diagnosed clinically, not with the use of manometer readings.

Current study found 95% prevalence of local envenoming including local pain, swelling, blistering, necrosis and local lymphadenopathy. One study points out a 100% of local envenoming found in all children [7]. In adult HNPV bites, this was 82–100% [2–5]. Dry bites, in present study were 5% whereas in adults, it ranges 5–9% [2–4]. Previous literature describes that envenoming in children can be more severe than adults and they have higher morbidity and mortality [15,23–25]. In addition to mortality, long term effects of amputations and post-traumatic stress disorder account for a great burden of snake envenoming in children [26]. But, in our study any child did not develop severe life-threatening complications and almost all fully recovered following HNPV bites. A previous adult study revealed eosinophilia as the key haematological parameter following HNPV bites [4]. But, in current paediatric study, it was the leucocytosis that is predominant and eosinophilia was observed only in 27%.

Neutrophil leukocytosis is a common observation in experimental animals in response to snake venoms [27].

Envenoming in children is generally severe due to the lower volume of distribution of venom. Additionally, they prone to develop potential for life long permanent sequelae from tissue damage due to necrosis and psychological sequelae. Children should be specially addressed when planning preventive strategies of snakebites because it disproportionally affects children living in impoverished rural communities [28]. Ideally, this can be achieved by education about snakebites in rural schools.

## Limitations

We encountered difficulty in pain assessment (mild, moderate and severe) as the one of main local envenoming effects in HNPV bites in children below the age of 3 years (n = 8).

## Conclusions

Hump-nosed pit viper bites mostly cause local effects and rarely systemic envenoming in children. Compartment syndrome is common in children following their bites and these children should be closely monitored in order to detect acute compartment syndrome early and to prevent permanent loss of limbs.

## Acknowledgments

We are thankful to the staff of paediatric unit and Consultant Paediatritians (Dr. Saman Abeywardhana, Dr. Lal Rathnasiri, Dr. Ananda Wijekoon, Dr.K.P.C. Pushpakumara), Teaching Hospital Ratnapura- Sri Lanka. Prof. R.P.V.J Rajapakse (Department of Veterinary Pathobiology, Faculty of Veterinary Medicine & Animal Science) and Prof. W.D.S.J.Wickramasinghe (Department of Parasitology, Faculty of Medicine) of University of Peradeniya were acknowledged for their constructive comments.

## Author Contributions

**Conceptualization:** R. M. M. K. Namal Rathnayaka, S. A. M. Kularatne.

**Data curation:** R. M. M. K. Namal Rathnayaka, P. E. Anusha Nishanthi Ranathunga.

**Formal analysis:** R. M. M. K. Namal Rathnayaka, P. E. Anusha Nishanthi Ranathunga.

**Investigation:** R. M. M. K. Namal Rathnayaka, P. E. Anusha Nishanthi Ranathunga.

**Methodology:** R. M. M. K. Namal Rathnayaka, S. A. M. Kularatne.

**Project administration:** R. M. M. K. Namal Rathnayaka, P. E. Anusha Nishanthi Ranathunga.

**Resources:** P. E. Anusha Nishanthi Ranathunga.

**Software:** R. M. M. K. Namal Rathnayaka, P. E. Anusha Nishanthi Ranathunga.

**Supervision:** S. A. M. Kularatne.

**Writing – original draft:** R. M. M. K. Namal Rathnayaka, P. E. Anusha Nishanthi Ranathunga, S. A. M. Kularatne.

**Writing – review & editing:** R. M. M. K. Namal Rathnayaka, P. E. Anusha Nishanthi Ranathunga, S. A. M. Kularatne.

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
