## [Decision Letter · Decision Letter 0]

26 Sep 2022

Dear Dr. Rathnayaka,

Thank you very much for submitting your manuscript "Epidemiological and clinical features of hump-nosed pit viper (Hypnale hypnale and H. zara) envenoming in children" for consideration at PLOS Neglected Tropical Diseases. As with all papers reviewed by the journal, your manuscript was reviewed by members of the editorial board and by several independent reviewers. The reviewers appreciated the attention to an important topic. Based on the reviews, we are likely to accept this manuscript for publication, providing that you modify the manuscript according to the review recommendations. 

In your response, please ensure you consider and include responses to points raised in the attachment as well as the comments listed below. 

Sincerely,

Stuart Robert Ainsworth

Academic Editor

Wuelton Monteiro

Section Editor

Reviewer's Responses to Questions

**Key Review Criteria Required for Acceptance?**

**Methods**

-Are the objectives of the study clearly articulated with a clear testable hypothesis stated?

-Is the study design appropriate to address the stated objectives?

-Is the population clearly described and appropriate for the hypothesis being tested?

-Is the sample size sufficient to ensure adequate power to address the hypothesis being tested?

-Were correct statistical analysis used to support conclusions?

-Are there concerns about ethical or regulatory requirements being met?

Reviewer #1: This is a prospective observational study.

The authors did not mention whether patients were followed for lab evidences of consumption coagulopathy at specific time intervals. As the 'bite to presentation' time was short, only admission profile will miss the developing coagulopathy. This is of more concern for the patients undergoing surgery.

Statistical design is ok.

Reviewer #2: -Are the objectives of the study clearly articulated with a clear testable hypothesis stated?

Since it is prospective and descriptive, the study meets the aims stated.

-Is the study design appropriate to address the stated objectives?

Yes.

-Is the population clearly described and appropriate for the hypothesis being tested?

yes.

-Is the sample size sufficient to ensure adequate power to address the hypothesis being tested?

As a descriptive study the number of cases is adequate.

-Were correct statistical analysis used to support conclusions?

yes

-Are there concerns about ethical or regulatory requirements being met?

Consent as required has been stated to have been taken.

**Results**

-Does the analysis presented match the analysis plan?

-Are the results clearly and completely presented?

-Are the figures (Tables, Images) of sufficient quality for clarity?

Reviewer #1: Needs correction of grammatical mistakes.

Few tables need corrections (stated below).

Reviewer #2: -Does the analysis presented match the analysis plan?

yes

-Are the results clearly and completely presented?

yes

-Are the figures (Tables, Images) of sufficient quality for clarity

yes

**Conclusions**

-Are the conclusions supported by the data presented?

-Are the limitations of analysis clearly described?

-Do the authors discuss how these data can be helpful to advance our understanding of the topic under study?

-Is public health relevance addressed?

Reviewer #1: Limitations and conclusions are not stated separately.

Reviewer #2: -Does the analysis presented match the analysis plan?

yes

-Are the results clearly and completely presented?

yes

-Are the figures (Tables, Images) of sufficient quality for clarity

yes

**Editorial and Data Presentation Modifications?**

Reviewer #1: (No Response)

Reviewer #2: minor revision

**Summary and General Comments**

Reviewer #1: This study tried to address the issue specifically in paediatrics age group to highlight the problem.

Reviewer #2: The study is relevant and simple. Yes, paediatric envenomations by the Hypnale spp are not well recorded. The quality of english could improve but it is generally clear. The process of identification of patients transferred in with identification of the snake by non-experts is a potential grey area due to misidentification which in turn skews data and its interpretation given that there are at least two species in question in this study. This may be a limitation of the study.

PLOS authors have the option to publish the peer review history of their article (what does this mean?). If published, this will include your full peer review and any attached files.

Reviewer #1: No

Reviewer #2: Yes: Freston Marc Sirur

Figure Files:

Data Requirements:

Reproducibility:

References

---

## [Decision Letter · Decision Letter 1]

9 Dec 2022

Dear Dr. Rathnayaka,

We are pleased to inform you that your manuscript 'Epidemiological and clinical features of hump-nosed pit viper (Hypnale hypnale and Hypnale zara) envenoming in children' has been provisionally accepted for publication in PLOS Neglected Tropical Diseases.

Best regards,

Stuart Robert Ainsworth

Academic Editor

Wuelton Monteiro

Section Editor

Reviewer's Responses to Questions

**Key Review Criteria Required for Acceptance?**

**Methods**

-Are the objectives of the study clearly articulated with a clear testable hypothesis stated?

-Is the study design appropriate to address the stated objectives?

-Is the population clearly described and appropriate for the hypothesis being tested?

-Is the sample size sufficient to ensure adequate power to address the hypothesis being tested?

-Were correct statistical analysis used to support conclusions?

-Are there concerns about ethical or regulatory requirements being met?

Reviewer #1: (No Response)

**Results**

-Does the analysis presented match the analysis plan?

-Are the results clearly and completely presented?

-Are the figures (Tables, Images) of sufficient quality for clarity?

Reviewer #1: (No Response)

**Conclusions**

-Are the conclusions supported by the data presented?

-Are the limitations of analysis clearly described?

-Do the authors discuss how these data can be helpful to advance our understanding of the topic under study?

-Is public health relevance addressed?

Reviewer #1: (No Response)

**Editorial and Data Presentation Modifications?**

Reviewer #1: (No Response)

**Summary and General Comments**

Reviewer #1: Can be accepted for publication.

PLOS authors have the option to publish the peer review history of their article (what does this mean?). If published, this will include your full peer review and any attached files.

Reviewer #1: No

---

## [Editor Report · Acceptance letter]

19 Dec 2022

Dear Dr. Rathnayaka,

We are delighted to inform you that your manuscript, "Epidemiological and clinical features of hump-nosed pit viper (Hypnale hypnale and Hypnale zara) envenoming in children," has been formally accepted for publication in PLOS Neglected Tropical Diseases.

Best regards,

Shaden Kamhawi

co-Editor-in-Chief

Paul Brindley

co-Editor-in-Chief
